# How to judge a competition:
# Fairly judging a competition or assessing benchmark results

**Adrien Pavão**                                                    PAVAO@MLCHALLENGES.COM
*LISN, CNRS*
*Université Paris-Saclay*
*France*

**Reviewed on OpenReview:** *https://openreview.net/forum?id=JLsZ9iNdhs*

## Abstract

This chapter addresses how to minimize randomness in competition or benchmark judging. We discuss scoring metrics, size of test data, error bars, splitting into phases, and score aggregation methods. Our approach blends theoretical insights with practical guidelines, aiming to provide a clear framework for effective decision making and reduced uncertainty.

**Keywords:** evaluation, metric, ranking, error bars, staged competitions

## 1 Introduction

Machine learning competitions, in many ways, resemble sport events. From the perspective of the organizers, much like in sports, there's a pursuit to rank participants fairly based on their skillset and adherence to a specific set of rules. One of the primary objectives in these scientific competitions is not just to crown a winner, but to address a particular problem or answer a scientific question. The emphasis lies in judging the participants justly and, even more crucially, in assessing the merit of their proposed solutions—the models—to make noteworthy advancements in problem-solving. It's vital to design tasks and evaluate contributions in a manner that fosters competitions with substantive outcomes rather than ones that can be exploited, intentionally or otherwise. A well-structured scientific competition can serve various purposes, such as stimulating research in a field or promoting a specific research line.

Beyond the goal of generating significant, reproducible, and universal results, there's a legal aspect to consider. In many jurisdictions, gambling is stringently regulated. As competition organizers, it's essential to distinguish these contests from games of chance. Often, the competition rules explicitly state: "this is a skill-based contest in which chance plays no role." However, this isn't always entirely accurate. Indeed, the cash prize might be awarded to a winner whose score is not significantly distinct from other competitors, as illustrated by the Wheat Detection Challenge (David et al., 2020), later in this chapter.

This chapter explores key aspects of organizing such competitions, including the selection of the scoring metric (Section 2), determining the statistical significance of results (Section 3), and addressing the challenge of score aggregation across various criteria (Section 4). For clarity, the structure of the chapter is illustrated by the Figure 1.

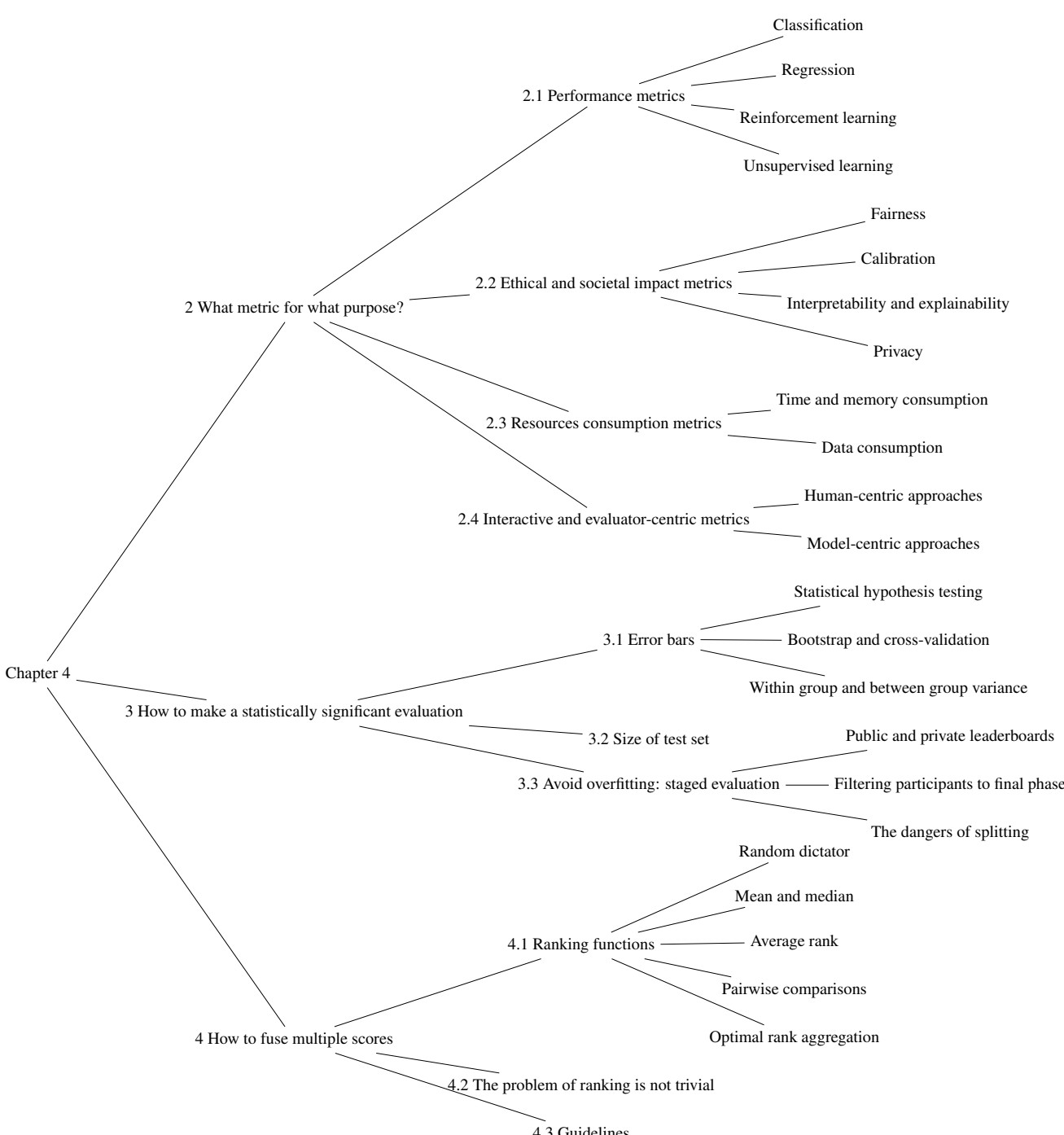

Figure 1: Structure of the chapter 4.

## 2 What metric for what purpose?

The evaluation metric is, obviously, the core of a challenge: it produces the value that everybody is trying to optimize. This is why the choice of the metric is one of the most important part of the definition of a problem. We have observed that the final ranking of methods is usually very sensitive to the choice of metric (Caruana and Niculescu-Mizil, 2004). This is why this choice needs to be made carefully: an unsuitable metric results in unsuitable solutions. Let's take some examples of such errors of design.

So, how to choose a metric that fits the problem? The variety of metrics that have been proposed in the literature is so large that is hard to find one's way. Thus, we present here the most used metrics in several area of machine learning and explain their main qualities and shortcomings. General view and survey of evaluation methods are provided by survey (Raschka, 2018; Hernández-Orallo et al., 2012).

In the following, we distinguish between performance metrics (e.g. accuracy), ethical and societal impact metrics (e.g. measure of fairness), resources consumption metrics (e.g. time consumption) and evaluator-centric metrics (e.g. human evaluation). Note that many classification, regression and clustering metrics are implemented[1] in the famous machine learning Python package Scikit-Learn (Pedregosa et al., 2011).

### 2.1 Performance metrics

In this section, we describe metrics commonly used as primary objective in classical machine learning problems: classification, regression, reinforcement learning and unsupervised learning.

CLASSIFICATION

A prediction task is called a *classification problem* when the possible outcomes to predict are grouped in different classes (Grandini et al., 2020). The simplest setting involves only two classes (binary classification, reviewed by Berrar (2019); Canbek et al. (2017)); classification tasks involving more than two classes are called *multi-class classification*. For instance, the classical problem of handwritten digits recognition (LeCun and Cortes, 2005) is a multi-class classification. In *multi-label classification*, each data point can be classified into several classes at the same time. The goal is to use available data called $X$ to obtain the best prediction $\hat{Y}$ of the outcome variable $Y$. In multi-class classification, $Y$ and $\hat{Y}$ can be seen as two discrete random variables that assume values in $\{1, \ldots, K\}$ where each number represents a different class, and $K$ is the number of distinct classes.

When classification algorithms output the probability that a sample from $X$ belongs to a given class; a classification rule is then employed to assign a single class. In binary classification, a threshold can be used to decide the predicted class. In the multi-class case, there are various possibilities, the most employed technique being selecting the highest probability value, commonly computed using the softmax function (Grandini et al., 2020).

To give a general overview of the resemblance between classification metrics, we conducted an experiment to empirically evaluate the correlation between common classification scoring metrics: we computed rankings of the final models from AutoDL Challenge, independently on all 66 datasets formatted for this competitions, and compared these rankings using Euclidean distance. The results are presented graphically in Figure 2, scaled in a two-dimensional plot. *Jaccard score* and *F1-score*

---

1. https://scikit-learn.org/stable/modules/model_evaluation.html

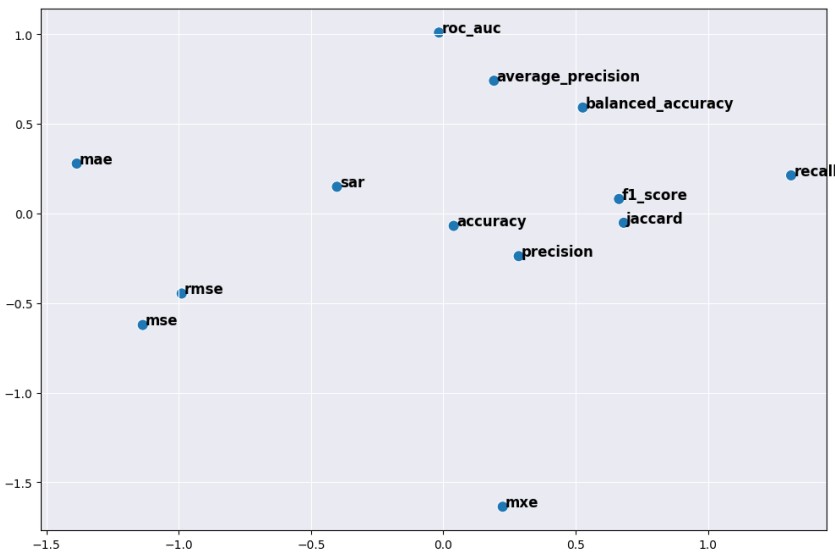

Figure 2: Multidimensional scale (MDS) plot illustrating the degree of correlation between scoring metrics in a 2D space. The metrics are compared by computing Euclidean distance between the rankings they produced. This experiment was performed on the classification tasks and models from AutoDL Challenge (Liu et al., 2021).

are confirmed to be very close[2]. The *SAR* metric (Caruana and Niculescu-Mizil, 2004), an average between *accuracy*, *AUC* and *root mean squared error* (using $1 - RMSE$), is well centralized, as it was designed to be. Interestingly, *balanced accuracy* seems to be centralized between *accuracy* and *AUC*. Loss functions, such as *mean absolute error* (MAE) and *mean squared error* (MSE), are clearly distinct from classification metrics.

Selecting the right scoring metric for a classification task is important as it directly impacts the evaluation and interpretation of the model's performance. The main points we identified as relevant to guide the selection process are the type of problem, the class balance and the real world objectives.

**Problem type**: Consider the type of problem. Most classification metrics are defined as binary classification metrics; however, they can be used to score multi-class problems, by breaking the problem down into multiple binary problems. The problems can be broken down using either *One-vs-One* (OVO) or *One-vs-Rest* (OVR) approaches. In the OVO approach, the pairwise score of all pairs of classes is computed. In the OVR approach, the scores for each class is computed separately, treating each class as the positive class and all other classes as the negative class. In both cases, the problem is broken down into a series of binary problems, and the final score is obtained by averaging all the scores, either using a simple average, or a weighted average. The OVO approach is computationally expensive, as the number of scores to compute is $\frac{K(K-1)}{2}$ with $K$ being the number of different classes. For this reason, the OVR approach is mostly used in practice, needing only $K$ computations, one for each class. We consider only the OVR approach for the rest of this section.

---

2. https://stats.stackexchange.com/questions/511395/are-jaccard-score-and-f1-score-monotonically-512378#512378

Beyond its computational benefits, the OVR approach is favored due to its simplicity, making it more interpretable for non-experts. It scales linearly as the number of classes grows, contrasting with the exponential complexity of the OVO method. Furthermore, its prominence in modern machine learning tools, given the optimized implementations in popular frameworks, emphasizes its relevance for real-world applications.

The simplest idea when it comes to score classifiers is to use *rates of success*. *Precision*, *recall* and *accuracy* are metrics that simply count the successes and failures of the classifier. Intuitively, *accuracy* is the likelihood that a randomly chosen sample will be correctly classified by the model. The fundamental component of this metric is the individual units in the dataset, each of which holds equal weight and contributes equally to the score. However, when considering classes instead of individuals, some classes may have a high number of units while others have only a few. In such cases, the larger classes will carry more weight compared to the smaller ones. When the dataset is imbalanced, meaning that most units belong to one particular class, *accuracy* may overlook significant classification errors for classes with fewer units as they are less significant compared to the larger classes.

**Class imbalance**: Consider the distribution of classes in the dataset. If there is a class imbalance, using accuracy as a metric might not provide an accurate picture of the model's performance. In such cases, metrics like *balanced accuracy* or *area under ROC curve* (AUC) are more appropriate. *Balanced accuracy* addresses this issue by giving each class equal impact on the score. This is simply done using a weighted average of each class accuracy, weighted by the proportion of the class in the test set. Despite having fewer units, smaller classes may have a disproportionately larger influence on the formula. When the dataset is relatively balanced, meaning the class sizes are similar, *accuracy* and *balanced accuracy* tend to produce similar results. The main distinction between the two metrics becomes apparent when the dataset exhibits an imbalanced distribution of classes.

**Real world objective**: Consider the real world impact of the problem and the cost of false positive and false negatives. For instance, in medical diagnosis, a false positive (e.g. a healthy person is diagnosed with a disease) can lead to unnecessary medical procedures and treatments, causing harm to the patient. On the other hand, a false negative (e.g. a sick person is not diagnosed) can result in a delay in treatment and a potentially fatal outcome. In this case, *recall* (the ability of the model to identify all positive cases) is a relevant metric. Another example: in the finance sector, false positives can lead to substantial revenue loss from incorrect trades or transactions, while false negatives might result in missed opportunities. In such contexts, *precision* (the model's capability to correctly identify positive instances) and *F1 score* (a harmonized metric between precision and recall) could be considered better choices for assessing the model's efficiency.

## Regression

The prediction task is called a regression problem when the outcome is a continuous numeric value, in contrast to a classification problem where the variable to predict falls into discrete categories. For instance, predicting temperatures or road traffic from support variables are regression tasks. A review of metrics for regression is given by (Botchkarev, 2018). Another survey concerns evaluation methods for time series forecasting (Cerqueira et al., 2020).

The main points we identified as relevant to guide the selection process are the type of problem, the scale of the target variable and the real world objectives.

**Problem type**: Consider the type of problem you are trying to solve. For example, for time-series forecasting problems, metrics like *mean absolute error* (MAE), *mean squared error* (MSE), and *root mean squared error* (RMSE) are relevant. For problems involving predicting counts, metrics like *mean absolute percentage error* (MAPE) and *symmetric mean absolute percentage error* (SMAPE) might be more appropriate.

**Scale of the target variable**: Consider the scale of the target variable. If the target variable is large, the absolute difference between the actual and predicted values will also be large, making *MAE*, *MSE* and *RMSE* less interpretable. In such cases *R-squared* (coefficient of determination) might be more appropriate metrics. *R-squared* has no units, can be compared among different tasks, and has an intuitive interpretation.

**Real world objective**: Once again, consider the real world problems. For instance, if the problem involves predicting stock prices, the magnitude of the error is more important than the direction of the error. In such cases, *RMSE* or *MSE* might be appropriate metrics. Another example: in the field of climate modeling, the right choice of scoring metric vary depending on the problem. If the goal is to predict global temperatures, metrics like *mean absolute error* (MAE) could be used to evaluate the performance of the model. However, in predicting regional precipitation patterns, metrics like *R-squared* or *explained variance score* might be used to evaluate the ability of the model to capture the complex spatial patterns of precipitation.

REINFORCEMENT LEARNING

Reinforcement learning (RL) consists, for an autonomous agent (e.g. robot), in learning what actions to take, based on experiences, in order to optimize a quantitative reward over time. The agent is immersed in an environment and makes its decisions based on its current state. In return, the environment provides the agent with a reward, which can be either positive or negative. The agent seeks, through iterated experiments, an optimal strategy or *policy*, which is a function that associates the current state with the action to be performed, to maximize the sum of rewards over time. This setting makes the scoring procedure particularly problem-specific and prone to design flaws.

In reinforcement learning, overfitting occurs when the agent becomes too specialized to the conditions of the training environment and is unable to generalize to unseen situations. This is a common problem in RL because the training and the evaluation processes are usually conducted on the same environment, rather than in two separate environments. This creates a situation where the agent can memorize the optimal actions in the training environment without truly understanding the underlying dynamics. As a result, the agent's performance may appear to be much better than it actually is, leading to a biased evaluation. This can be a serious issue in RL, as it undermines the validity of the evaluation process and may result in the selection of sub-optimal algorithms or policies. To mitigate this problem, it is important to separate the training and testing environments and use different metrics and simulators for evaluation. This helps to unbias the evaluation process and to ensure that the results accurately reflect the true performance of the agent.

Some research studying the evaluation of RL algorithms show that behavioral metrics play a crucial role in determining the quality of a state representation, and in learning an optimal representation (Jordan et al., 2020; Lan et al., 2021). Existing methods, such as *approximate abstractions* and *equivalence relations*, aiming at reducing the size of the state or action space by aggregating similar states, are not effective for continuous-state reinforcement learning problems, due to their inability to maintain the continuity of common RL functions and their tendency to generate overly

detailed representations that lack generalization. A behavioral metric in reinforcement learning is a measure of an agent's performance in an environment, based on its actions and observed rewards. It can be used to evaluate and compare different reinforcement learning algorithms or policies. Examples include *average reward per episode*, *success rate*, and *convergence speed*.

According to Henderson et al. (2018), multiple trials with different random seeds are necessary to compare performance, due to high variance. To ensure reproducibility, it is crucial to report all hyperparameters, implementation details, experimental setup, and evaluation methods for both baseline comparisons and novel work.

### UNSUPERVISED LEARNING

Numerous tasks in machine learning lack a definitive ground truth for evaluating solutions. Termed as unsupervised learning, this category includes a diverse range of tasks, from clustering and dimensionality reduction to data modeling, generation, and feature extraction. It also covers areas with a more subjective nature, like automatic music composition, identifying molecules that bind to COVID-19, and text summarization.

It is particularly challenging to design competitions for such unsupervised learning tasks, due to the following reasons: the lack of ground truth, the diversity of solutions, and the subjectivity in evaluation. Indeed, unlike supervised learning, unsupervised learning often lacks clear ground truth, making it difficult to evaluate the results objectively. Also, unsupervised learning models can often produce diverse solutions that are equally valid, making it challenging to select a single best solution. Finally, the evaluation of unsupervised learning solutions can be subjective as it depends on the understanding and interpretation of the evaluators.

In the case of distribution modelling and clustering, some clear performance metrics can be identified. Prior research investigate the evaluation metrics for unsupervised learning (Palacio-Niño and Berzal, 2019) and clustering (Ben-Hur et al., 2002; von Luxburg et al., 2012). Typically, when the goal in unsupervised learning is to **learn the underlying data distribution**, various loss functions can be employed depending on the specific type of model:

- *Maximum Likelihood Estimation* (MLE): In probabilistic models, the goal is often to maximize the likelihood of the observed data.

- *Kullback-Leibler (KL) Divergence* (Kullback and Leibler, 1951): Measures the difference between two probability distributions. It is often used with Variational Autoencoders (VAEs) (Kingma and Welling, 2019) where one tries to minimize the divergence between the learned distribution and the true data distribution.

- *Reconstruction Loss*: In models like autoencoders (Liou et al., 2014), the objective is to reconstruct the input data from a compressed representation. The loss measures the difference between the original input and its reconstruction.

- *Wasserstein Distance* (Earth Mover's Distance) (Villani, 2009): Used in Wasserstein Generative Adversarial Networks (WGANs) (Arjovsky et al., 2017) to measure the distance between the generated distribution and the true data distribution.

To assess the performance of **clustering** methods, when no ground truth is available, the following metrics and techniques can be used:

- *Inertia* (Sum of Squared Errors): Inertia measures the sum of squared distances between each data point and its closest cluster center. Lower inertia values indicate tighter clusters and better performance.

- *Silhouette Score* (Rousseeuw, 1987): This metric computes the average silhouette value for all data points, measuring how similar a data point is to its own cluster compared to other clusters. A silhouette score close to 1 indicates a well-partitioned dataset, whereas a score close to -1 indicates poor clustering.

- *Davies-Bouldin Index* (Davies and Bouldin, 1979): This index measures the ratio of within-cluster scatter to between-cluster separation. Lower values of Davies-Bouldin Index indicate better clustering performance.

- *Calinski-Harabasz Index* (Caliński and Harabasz, 1974): This index evaluates clustering by comparing the ratio of between-cluster dispersion to within-cluster dispersion. Higher values of the Calinski-Harabasz Index suggest better clustering performance.

Although these metrics are valuable for tasks like distribution learning and clustering, they fall short in assessing the performance of machine learning models in areas like text summarization or artistic creation. In these scenarios, it is challenging to score the success. Interesting and effective methods include human evaluation, employing other machine learning models as evaluators, and using interactive adversarial frameworks.

Indeed, **human evaluation** can play a crucial role in assessing the performance of unsupervised learning algorithms when traditional quantitative metrics may not fully capture the desired outcomes, or when ground truth labels are not available. Human evaluation can provide valuable qualitative insights and help ensure that the resulting patterns or structures discovered by the algorithms align with human intuition and domain knowledge. To perform human evaluation effectively, it is essential to establish clear guidelines for the evaluators, provide proper training, and, when possible, recruit multiple evaluators to increase the reliability of the assessment. Human evaluation can be time-consuming and resource-intensive, so it is often used in combination with quantitative metrics to balance the efficiency and quality of evaluation. This approach is further discussed in Section 2.4.

Another interesting technique, that may become more popular in the future, is to use a **model used as a metric**. For instance, a classifier trained on discriminating between two distributions (e.g. fake and real) can be used to evaluate the performance of generative models. This can be compared to the functioning of Generative Adversarial Networks (GAN) (Goodfellow et al., 2014), where the output of a binary classifier is used to guide the learning of a generative model. When using another machine learning model as a metric for unsupervised learning, it is important to remember that the evaluation depends on the performance of the supervised model and the quality of the labeled data. Therefore, the results should be interpreted with caution, as the evaluation might be biased or limited by the chosen supervised model or the available data. This approach is discussed in Section 2.4.

Finally, adversarial challenges, where the solutions proposed by the participants are then used as input data in the next phase, is potentially a good way of organizing challenges on unsupervised learning tasks. Adversarial challenges design are explored in details in Chapter 12.

More generally, to overcome these challenges and organize unsupervised learning competitions and benchmarks, it is essential to articulate the problem and describe the data comprehensively.

Evaluation methods must be defined considering the missing ground truth. Promoting collaboration among participants can diversify the solutions. Involving domain experts in the evaluation can enhance the result objectivity. Keeping participants informed about competition progress and being open to adjustments can optimize the process. Lastly, providing evaluations using diverse metrics can help participants gauge the pros and cons of their methods. By following these steps, competitions can be designed to effectively evaluate the solutions to an unsupervised learning task while overcoming the challenges of lack of ground truth, diversity of solutions, and subjectivity in evaluation.

## 2.2 Ethical and societal impact metrics

Traditional evaluation metrics like *accuracy*, *precision*, and *recall* have long held the spotlight, but there is so much more to consider when measuring a model's real-world impact. In this section, we dive into the lesser-known, unconventional metrics that are reshaping the way we assess machine learning models. From fairness and privacy to interpretability and calibration, these innovative evaluation techniques change how we think about model performance and pave the way for a more responsible, holistic approach to machine learning. Indeed, the performance metrics we have reviewed in this chapter so far reflect only one aspect of the performance of the models: their predictive abilities. Yet, in many applications, our concerns extend to other aspects, like the trustworthiness of the algorithms. This is particularly true in sensitive applications, where an algorithmic decision could mean life or death.

### FAIRNESS

Even if the mathematical definition of machine learning models does not necessarily contains unfair or biased elements, trained models can be unfair, depending on the quality of their input data or their training procedure. A model trained on biased data may not only lead to unfair and inaccurate predictions, but also significantly disadvantage certain subgroups, and lead to unfairness. In other words, the notion of fairness of models describes the fact that models can behave differently on some subgroups of the data. The issue is especially significant when it pertains to demographic groups, typically defined by factors such as gender, age, ethnicity, or religious beliefs. As machine learning is increasingly applied in society, this problem is getting more attention and research, and is subject to debate (Benz et al., 2020; Vasileva, 2020; Chouldechova and Roth, 2018; Chen et al., 2018; Boratto et al., 2021). Some interesting ways to quantify fairness include:

**Demographic Parity**: This measure checks if the positive classification rate is equal across different demographic groups. The formula is as follows:

$$DemographicParity : P(\hat{Y} = 1 | A = 0) = P(\hat{Y} = 1 | A = 1)$$

where $A$ is a protected attribute (such as race or gender), $Y$ is the target variable (such as approval or denial) and where $\hat{Y}$ is the predicted value of $Y$. Demographic parity is a condition to be achieved: the predictions should be statistically independent of the protected attributes.

**Statistical Parity Difference**: Related to the demographic parity, it measures the difference between positive classification rate across different demographic groups. The formula is:

$$StatisticalParityDifference = P(\hat{Y} = 1 | A = 0) - P(\hat{Y} = 1 | A = 1)$$

**Disparate impact**: It calculates the ratio of the positive classification rate for a protected group to the positive classification rate for another group. It is similar to the *statistical parity difference*, but it is a ratio instead of a difference:

$$DisparateImpact = \frac{P(\hat{Y} = 1|A = 0)}{P(\hat{Y} = 1|A = 1)}$$

A value of 1 indicates that the positive classification rate is the same for both groups, suggesting fairness. A value greater than 1 indicates a higher positive classification rate for the group with $A = 0$, while a value less than 1 suggests a higher positive classification rate for the group with $A = 1$. However, it is important to note that disparate impact is a limited measure of fairness and should not be used on its own. There may be cases where a higher positive classification rate for one group is justifiable, for example if the group is underrepresented in the training data. Additionally, disparate impact does not consider other factors such as false positive and false negative rates, which may provide a more comprehensive view of fairness.

**Equal Opportunity**: This metric checks if the true positive rate is equal across different demographic groups. The formula is:

$$EqualOpportunity : P(\hat{Y} = 1|Y = 1, A = 0) = P(\hat{Y} = 1|Y = 1, A = 1)$$

As for *demographic parity*, it is a condition to be achieved.

**Equal Opportunity Difference**: This metric measures if the true positive rate is equal across different demographic groups. The formula is:

$$EqualOpportunityDifference = P(\hat{Y} = 1|Y = 1, A = 0) - P(\hat{Y} = 1|Y = 1, A = 1)$$

The same idea can be applied to false positive rates.

These are just a few of the metrics that can be used to quantify fairness in classification tasks. It is important to note that fairness is a complex issue, and these metrics should not be used in isolation. Instead, they should be considered in the context of the specific problem and the desired outcome.

## CALIBRATION

Classifiers usually return probabilities to indicate the confidence levels across different classes. However, the question arises whether these confidence levels accurately reflect the classifier's actual performance. As defined by Naeini et al. (2015); Guo et al. (2017), the notion of miscalibration represents the difference in expectation between the confidence level (or probability) returned by the algorithm, and the actual performance obtained. In other words, calibration measurement answers the following question: is the confidence of the algorithm about its own predictions correct? Promoting well calibrated models is important in potentially dangerous decision making problems, such as disease detection or mushroom identification. The importance of calibration measurement lies in the fact that it is essential to have a clear understanding of the confidence level that the algorithm has in its own predictions. A well-calibrated algorithm will produce confidence levels that accurately reflect the likelihood of a prediction being correct. In contrast, a miscalibrated algorithm

will either over or under estimate its confidence in its predictions, leading to incorrect or unreliable outcomes. In applications where the consequences of incorrect decisions can be severe, it is of utmost importance to have a well-calibrated algorithm. Misclassification of a disease can lead to incorrect medical treatment and harm to the patient. Similarly, misidentification of a mushroom can result in serious health consequences. In these scenarios, well-calibrated models can help ensure that the decisions are made based on reliable predictions.

The calibration can be estimated using the **Expected Calibration Error** (ECE): this score measures the difference between the average predicted probability and the accuracy (i.e., the proportion of positive samples) in bins of predicted probability. The formula for the ECE is given by:

$$ECE = \sum_{m=1}^{M} \frac{|B_m|}{n} |acc_m - conf_m|$$

where $M$ is the number of bins, $B_m$ is the set of samples in the $m^{th}$ bin, $n$ is the total number of samples in the test data, $acc_m$ is the accuracy of the $m^{th}$ bin, and $conf_m$ is the average predicted probability in the m-th bin.

When computing the calibration, we derive the performance prediction directly from the model's output. Another interesting possibility is to ask the participants to provide an estimation of the generalization score of their method. Indeed, we can make a connection between the calibration and the prediction of generalization error, more commonly estimated by a separated method. The Performance Prediction Challenge (Guyon et al., 2006) focused on this problem.

## INTERPRETABILITY AND EXPLAINABILITY

Given the complexities of machine learning models, the assessment of their interpretability and explainability emerges as a considerable challenge. While both concepts are crucial for ensuring trust and understanding in model predictions, especially in critical applications, measuring them accurately is difficult, especially when models vary widely in their structures and underlying mechanisms. Interpretability and explainability are related but distinct concepts in machine learning.

**Interpretability** refers to the degree to which a human can understand the cause of a model's predictions. It refers to the ability to understand the internal workings of the model and how it arrived at its decisions.

**Explainability** refers to the ability to provide a human-understandable explanation of the model's decision making process. It is concerned with the presentation of the reasons behind the predictions to humans in a understandable form, e.g. through feature importance.

In summary, interpretability focuses on the transparency of the model itself, while explainability focuses on the communication of the model's behavior to a human audience. A wide survey on interpretability is proposed by Carvalho et al. (2019). They stressed out how interpretability is greatly valuable in one hand, but hard to define in the other hand. Another way to explain algorithms, automatically, is the sensitivity analysis (Iooss et al., 2022). Sensitivity analysis is a technique used to determine how changes in input variables of a model or system affect the output or outcomes of interest.

Past competitions have been exploring the development and evaluation of explainable models, such as the *Job Candidate Screening Challenge* (Escalante et al., 2017, 2018). It is a challenge of first impressions and apparent personality analysis, on audio-visual data. The candidate models

have to predict apparent traits of people[3] (e.g. friendly or reserved, imaginative or practical) from short videos, with a focus on the explanatory power of techniques: solutions have to "explain" why a given decision was made. To this end, participants had to provide a textual description that explains the decision (i.e. the prediction) made. Optionally, participants could also submit a visual description to enrich and improve clarity and explainability. Performance was evaluated in terms of the creativity of participants and the explanatory effectiveness of the descriptions. For this evaluation, a set of experts in the fields of psychological behavior analysis, recruitment, machine learning and computer vision was invited. We can note that, this way, the explainability component of the challenge requires qualitative evaluations and hence human effort.

It is worth noting that some models are interpretable by nature, such as logistic regression or decision tree. Some researchers make the point that there is a trade-off between interpretability and models' performance, especially for complex tasks, that seem to be requiring blackbox models – huge deep learning neural networks. However, Rudin (2019) argues that this "accuracy-interpretability trade-off" is an unfounded myth.

## PRIVACY

Privacy must typically be measured when the candidate algorithms are generative models, modelling a distribution of potentially confidential data. The goal in such a case is to use the generative models in order to create artificial data that reassemble sufficiently the real data to be used in actual applications, but not too much for private information to be leaked. A metric to estimate this trade-off is the **adversarial accuracy**, that we introduced in Yale et al. (2019). Here is its definition:

$$AA_{TS} = \frac{1}{2} \left( \frac{1}{n} \sum_{i=1}^{n} \mathbf{1} \left( d_{TS}(i) > d_{TT}(i) \right) + \frac{1}{n} \sum_{i=1}^{n} \mathbf{1} \left( d_{ST}(i) > d_{SS}(i) \right) \right)$$

where the indicator function $\mathbf{1}$ takes the value 1 if its argument is true and 0 otherwise, $T$ and $S$ are true and synthetic data respectively. $d$ is an arbitrary chosen distance function, such as the Euclidean distance. $d_{TS}(i)$ represents the distance between the $i^{th}$ point of $T$ and its closest neighbor from $S$. $d_{TT}(i)$ is the distance between this $i^{th}$ point and its closest neighbor in $T$. Subsequently, $d_{ST}(i)$ and $d_{SS}(i)$ compare the $i^{th}$ point of $S$ to its closest neighbors in $T$ and in $S$.

It is basically the accuracy of a 1-nearest-neighbor classifier, but the ideal score is not 1 (perfect classification accuracy) but 0.5. Indeed, a perfect score means that each generated data point has its closest neighbor in the real data, which means that the two distributions are overly similar. A score of 0 would mean that the two distributions are too different, thus the practical utility is low. Hence, a 0.5 score, where the closest neighbor of each data point can either be fake or real with the same probability, is what guarantees a good privacy. These principles are illustrated with a toy example in Figure 3. Alaa et al. (2022) proposes a similar approach.

One limitation of this method is that a proper measure of distance is needed. This is also a strength because it means that the method is general and can be applied in different fields, by selecting an adequate distance measure.

In the study of privacy, *differential privacy* and *membership inference attacks* are core concepts.

*Differential privacy* provides a robust framework to ensure that a trained model does not get substantially influenced by the inclusion or exclusion of a single data sample from the dataset. It employs a parameter $\varepsilon$ to quantify the privacy, with smaller $\varepsilon$ values meaning more privacy

---

3. The data was labeled by around 2500 annotators.

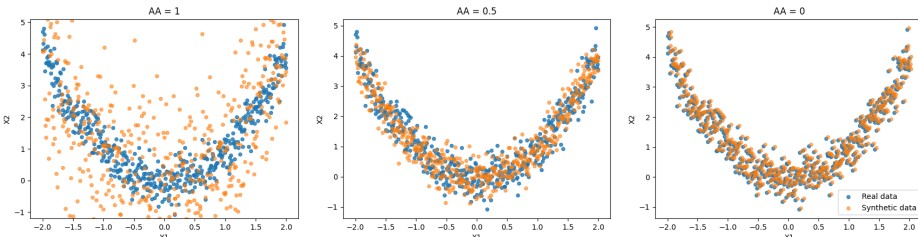

Figure 3: Adversarial Accuracy (AA) is the performance of an nearest neighbor classifier that distinguishes between real data vs synthetic data. The ideal value is $AA = 0.5$ (represented by the plot in the middle). This is a bi-variate ($X1$ and $X2$) illustrative example.

guarantees. It relates to the metric we proposed, which estimates the proximity between generated data and training data. While our approach evaluate the privacy preservation at the scale of the dataset, differential privacy focuses on individual-level privacy.

*Inference attacks* represent methods by which attackers deduce sensitive information using the models' output (or predictions). The adversarial accuracy metric intends, in some way, to measure the sensitivity of the model to these inference attacks. If generated data were nearly identical to training data, adversaries might be able to get sensitive information. By ensuring a degree of distance between original and generated data, the resistance against such attacks is improved.

Additionally, privacy concerns are not limited to synthetic data or generative models. Predictors too can be the target of privacy attacks, as highlighted by Pedersen et al. (2022).

## 2.3 Resources consumption metrics

Resource consumption metrics quantify the energy, time, and data utilized by models, thereby providing insights into their efficiency and sustainability.

### TIME AND MEMORY CONSUMPTION

It is useful to include the consumption of time, memory and energy of ML models in their evaluation and comparison. There are two main approaches to take these into account: **limit the resources** and **track the use of resources**. Both may imply in practice the use of code submission, as opposed to results submission. In code submission competitions, the participants submit their models which then get trained and tested on the servers, while in results submission competitions, the participants work locally and upload their predictions to the platform. Code submission is therefore advised if limiting or tracking the resource usage is part of the competition design.

The **training and inference time**, the **size of the model**, the **memory used** during the process or even the **energy consumption** are variables that can be limited by design or measured and shown on the leaderboard. Obviously, using the same hardware and evaluation conditions for all participants is needed in order to have a fair evaluation. The number of lines of code, or the number of characters, can also be used as an indicator of the **simplicity and practicability** of the solution. However, obviously, this indicator can be easily tricked by calling external packages and may need

a manual review. The simplest models that solve the task are preferable, for being less harmful for the environment, less costly, deployable in weaker devices and easier to interpret.

A model that can produce the same results in less time is more desirable, as it reduces the computational resources required and can lead to cost savings. This is especially important in light of the current ecological crisis, as reducing energy consumption in computing can have a significant impact on reducing the carbon footprint of technology. Additionally, models that are faster to train and make predictions are more scalable and can be deployed in real-time applications, further enhancing their utility. Thus, optimizing time consumption is a key factor in the development of efficient and environmentally sustainable machine learning models.

## ANYTIME LEARNING

Anytime learning refers to a learning paradigm in which a machine learning model or algorithm incrementally improves its performance as it receives more data or training time. The key aspect of anytime learning is that the model can produce meaningful results at any point during the learning process, with its performance generally improving as it acquires more data or spends more time on training. Anytime learning algorithms are particularly valuable in situations where resources, such as time or computation power, are limited, or when it is essential to provide real-time or near-real-time insights. These algorithms can be employed in various machine learning settings, including classification, regression, and reinforcement learning tasks.

To evaluate the score in this framework, one can compute the Area under the Learning Curve (ALC):

$$ALC = \int_{t_0}^{t_f} s(t)dt$$

where $s(t)$ is the performance score (obtained form a metric depending on the task) at timestamp $t$. $t_0$ and $t_f$ refers to the first and last timestamps, and should be fixed to allow a fair evaluation and comparison of the scores.

The time can be linear, or transformed at any scale:

$$ALC = \int_{t_0}^{t_f} s(t)d\tilde{t}(t)$$

where $\tilde{t}$ is the transform function. For instance, a logarithmic scale transform can be used, in order to give more importance to the first steps of training:

$$\tilde{t}(t) = \frac{\log(1+t)}{\log(1+t_f)}$$

The metric is depicted in Figure 4. Although both *Model 1* and *Model 2* converge to the same score, *Model 1* is favored in an anytime learning context due to its rapid performance improvement across epochs. This advantage is evident from the larger area under its learning curve.

Any-time learning can be linked to multi-fidelity. Multi-fidelity methods in machine learning refer to strategies that use various "fidelities" or qualities of data or models to speed up the learning process. For instance, one might use a simpler, faster-to-evaluate but less accurate model (low fidelity) to guide the learning process in the early stages, and a more complex, accurate but computationally intensive model (high fidelity) later on. The idea is to use less expensive resources to get

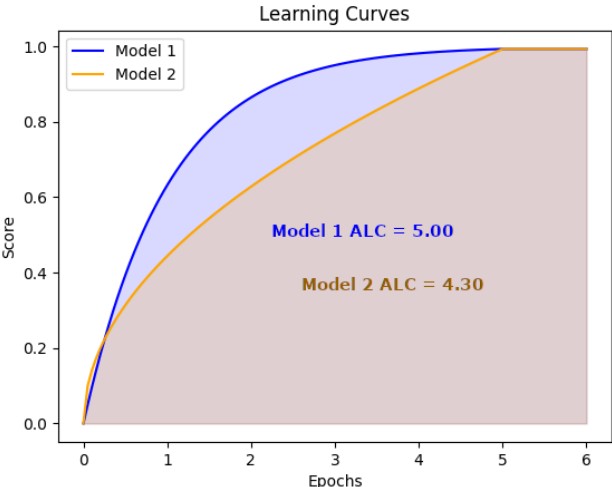

Figure 4: Example of learning curves for two models. While both *Model 1* and *Model 2* converge to the same score, *Model 1* boasts a larger area under the learning curve (ALC). Thus, in an anytime learning context, *Model 1* is the preferred choice.

an initial idea, and then refine it with more accurate but costlier methods. One could combine any-time learning and multi-fidelity methods to create a learning process that is both time-efficient and increasingly accurate. For example, in the early stages of an anytime learning algorithm, one might employ low-fidelity models or data to quickly get a "good enough" model. As time allows, the algorithm could then switch to using high-fidelity models or data to refine its understanding further. This way, one gets the best of both worlds: a quickly usable model, while aiming for high accuracy given more time. This combination could be particularly useful in scenarios where computational resources or time are constrained but where the model quality also needs to be maximized, such as real-time analytics, robotics, or complex simulations.

DATA CONSUMPTION

In machine learning, data is a key resource, and examining its consumption is relevant. The data input of models can be divided into two dimensions: the number of samples, and the number of features.

**Number of training samples.**

The amount of training data required is an interesting metric to consider when comparing differ-ent models. As the saying goes, "data is the new oil", but not every situation allows for the luxury of vast datasets. In many real-world applications, gathering sufficient labeled data can be time-consuming, expensive, or even impossible. Tracking and limiting data consumption is, therefore, an essential aspect of model evaluation.

Monitoring data consumption can help identify algorithms that perform well with limited data, making them more suitable for scenarios with small datasets. On the other hand, constraining the quantity of available training data can encourage the development of models that are more efficient

in learning. This is typically called few-shots learning, where meta-learning techniques, such as the $k$-shot $n$-way approach, are used. In this method, models are trained to quickly adapt to new tasks using only a limited number of examples $k$ from each class $n$. This $k$-shot $n$-way design was used in MetaDL competition (El Baz et al., 2021). By intentionally limiting data consumption, meta-learning promotes the development of models capable of generalizing better from smaller datasets, ultimately enhancing their utility and adaptability in diverse situations.

**Feature selection.**

Feature selection is the process of selecting a subset of relevant features, or variables, for use in model construction. Even if recent trends in deep learning tend to use the raw data without further preparation, feature selection is an essential aspect of machine learning that aims at selecting the most informative features for a given model. Proper feature selection can lead to simpler, more interpretable, and faster-performing models that may also have improved generalization.

Two primary methods exist to assess feature selection: **evaluation metrics** and **intrinsic metrics**. Evaluation metrics, also called the *wrapper approach*, assess a model's performance after feature selection, based on the assumption that effective model performances signify well-chosen features. The specific metrics used vary depending on the nature of the problem, such as classification or regression, as elaborated previously in this chapter. On the other hand, intrinsic metrics, also called the *filter approach*, evaluate the inherent quality or relevance of features without necessarily training a model. They act as heuristics of the fundamental information contained within each feature. Intrinsic metrics evaluate the significance of individual features without necessitating a full model. These include the *correlation coefficient*, which gauges the linear relationship between a feature and the target; *mutual information*, indicating the relevance of a feature based on how much it informs about another variable; *feature importance* from tree-based models such as decision trees and random forests; and *variance threshold*, where low-variance features, assumed less informative, might be discarded. While Kohavi and John (1997) advocates for the wrapper approach (evaluating trained algorithms), Tsamardinos and Aliferis (2003) argue that neither approach is inherently better, and that both the learner and the evaluation metric should be considered. Hybrid approaches, such as maximizing a performance score while minimizing the number of features, are efficient in balancing between model accuracy and model simplicity.

In the NIPS Feature Selection Challenge (Guyon et al., 2004; Guyon et al.), participants were ranked on the test set results using a score combining *Balanced Error Rate* (BER), the fraction of features selected $F_{feat}$, and the fraction of probes found in the feature set selected $F_{probe}$. The aim of feature selection is often to reduce the feature set's size without a significant loss in predictive performance. Hence, a lower $F_{feat}$ could be seen as favorable if the model's performance remains strong. "Probes" in the context of this challenge refer to "dummy" or "non-informative" features that were intentionally added to the dataset. These features don't have any relation or correlation with the target variable and are essentially noise. Thus, a good feature selection algorithm should ideally avoid selecting these probes, minimizing $F_{probes}$.

Briefly: they used the McNemar test (McNemar, 1947) to determine whether classifier $A$ is better than classifier $B$ according to the BER with 5% risk yielding to a score of 1 (better), 0 (don't know) or -1 (worse). Ties (zero score) were broken with $F_{feat}$ (if the relative difference in $F_{feat}$ was larger than 5%). Remaining ties were broken with $F_{probe}$. The overall score for each of the five datasets was the sum of the pairwise comparison scores.

These methods of feature selection, from intrinsic metrics to challenge-specific criteria such as those in the NIPS Feature Selection Challenge, play a role in optimizing machine learning models simplicity, performance and interpretability, and reducing their data consumption.

### 2.4 Interactive and evaluator-centric metrics

Evaluator-centric metrics represent a paradigm in machine learning evaluation where the benchmarking process actively involves specific evaluators, be they humans or models. Within this category, there is a distinction: human-centric approaches primarily leverage human judgment and perspectives, while model-centric approaches utilize predefined algorithms or models for evaluation.

#### HUMAN-CENTRIC APPROACHES

In addition to the quantitative evaluation metrics discussed previously, it is essential to consider more "human" evaluation techniques when assessing machine learning models. These approaches place emphasis on qualitative aspects and subjective interpretation, bringing a human touch to the evaluation process. For instance, in the case of text-to-image algorithms, manual inspection of generated images can help determine whether the outcomes are visually appealing, coherent, and contextually relevant. More generally, models that produce art, such as automatic music generators, benefit from manual evaluation. AI art competitions typically involve human evaluation through voting, such as the *Deep Art* competition of NeurIPS 2017[4]. IEEE CEC also hosts regular art competitions [5]. Similarly, large language models can be subjected to psychological or behavioral tests, where human evaluators rate the model's responses based on factors such as coherence, empathy, and ethical considerations. Such human-centric evaluation methods can reveal insights that purely numerical metrics might overlook, providing a more nuanced understanding of a model's strengths and weaknesses. It is important to distinguish between human evaluation and the comparison to human performance. While both are human-centric approaches, the latter specifically uses human abilities as a baseline for performance comparisons. A clear example of this approach is how the Generative Pre-trained Transformers (GPT) (OpenAI, 2023), the famous large language models, was tested using psychology tests (Uludag and Tong, 2023; Li et al., 2022), high-school tests (de Winter, 2023) and mathematics tests (Frieder et al., 2023). By integrating these human-oriented techniques into our evaluation toolbox, we can ensure that our machine learning models are not only effective in solving problems but also resonate with the multifaceted nature of human experiences.

#### MODEL-CENTRIC APPROACHES

A model can be used as a metric to evaluate the performance of other models, offering a dynamic and specialized approach to scoring complex tasks. This approach can be called model-centric, as opposed to human-centric approaches discussed in the previous section.

Typically, discriminative models can be used to assess the performance of generative models. Examples of this were given with the use of $k$-nearest neighbor adversarial accuracy to compute privacy, and the use of a classifier to score an image generation task. More generally, in this case, the

---

4. `https://nips.cc/Conferences/2017`
5. `https://sites.google.com/view/ieeecec2021ecmac/`

discriminant is trained to distinguish between real data from the target distribution and artificially generated data, thereby judging how "realistic" the generated data appears to be. This approach is similar to the learning framework of generative adversarial networks (Goodfellow et al., 2014). However, as underscored by the metric of adversarial accuracy for privacy, a generative model that completely deceives the discriminative model – in the sense that the discriminant gets an extremely low score – is an indication of privacy leakage of the training data. In other words, if the generative model performs exceptionally well within this adversarial framework, it raises concerns about its ability to generate general and original data. To avoid this, an "originality" or "privacy" metric should be invoked to measure the similarity, as mentioned in the Section 2.2. One distinct advantage of employing a discriminative model for performance measurement is its ability to output numerical values. Consequently, this form of performance assessment can easily be incorporated into a ranking system or any quantitative evaluation framework. An illustration of this protocol can be found in the Dog Image Generation challenge (Kan et al., 2019).

Another, more qualitative, way of measuring performance using models emerges through the use of language models. Large Language Models (LLMs) can serve as evaluators on various Natural Language Processing (NLP) tasks. For instance, in text summarization, an LLM can be employed to measure the semantic coherence and relevance of generated summaries by comparing them with the original text. The LLM could produce a likelihood score or even generate textual critiques to indicate how well the summary captures the essence of the source material. When it comes to explainability, an LLM can analyze the output explanations of complex models to assess their clarity and coherence. Naturally, as with any model-based metric, the initial prerequisite is to have confidence in the reliability of the evaluating model. More broadly, LLMs can act as judges for smaller models in a variety of NLP tasks. They can evaluate the quality of machine-generated translations, assess the sentiment consistency in chatbot dialogues, or even measure the relevance of answers generated by a question-answering system. These ideas were explored by the innovative competition *Auto-Survey Challenge 2023*[6] (Khuong and Rachmat, 2023). In this challenge, the participants propose AI agents capable of composing scientific survey papers and reviewing them. Such AI agents thus operate either as authors or reviewers. API calls to chatGPT were used to output the scores of *conclusion* (how well the conclusion highlights the main findings in the text) and *contribution* (relevance of the paper). To bridge between qualitative and quantitative outputs, the organizers asked the LLM to provide a number in Likert scale (Likert, 1932), for better differentiation between good and bad results (for instance, *1 - Strongly Disagree*, *2 - Disagree*, *3 - Neutral*, *4 - Agree*, *5 - Strongly Agree*). They also made clear and complete prompts, detailing how the characteristics should be evaluated.

Using a model as a metric offers many advantages and may even be essential for certain applications, but it also comes with its notable drawbacks and challenges. One of the immediate concerns is the additional layer of complexity and computational cost involved in using one model to evaluate another, which becomes particularly problematic when computational resources or time are limited. This issue is closely followed by questions regarding the reliability and consistency of the metric model itself. If the model employed as a metric has inherent weaknesses or biases, these could be transferred to the evaluation of the models being evaluated. The method's potential unreliability, complexity, and sensitivity to data and hyperparameters can result in difficult interpretations and risk misleading evaluations, especially in critical fields like healthcare and legal decision-making.

---

6. https://www.codabench.org/competitions/1145/

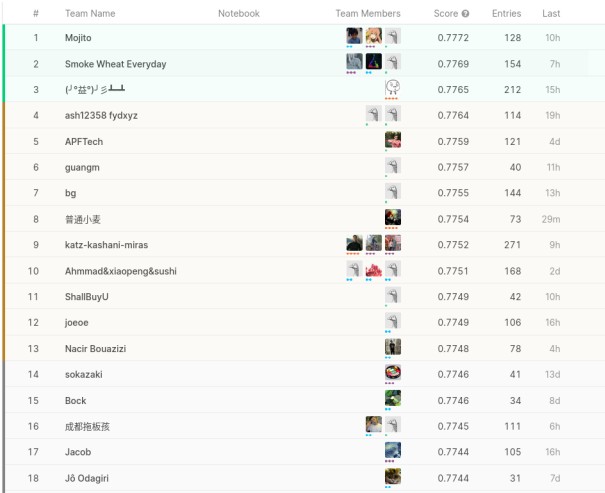

Figure 5: The (very tight) leaderboard from the Global Wheat Detection challenge (David et al., 2020) on Kaggle (Goldbloom and Hamner, 2010). Even if the scores are close, only the top-3 candidates share the $15,000 cash prize.

This vulnerability introduces the risk of "circular reasoning", particularly when the metric model is trained on similar data or shares architectural components with the model being evaluated, potentially leading to overly optimistic results. Typically, the organizers of the *Auto-Survey Challenge 2023* reported that chatGPT was usually overly optimistic and tended to grade its own work better than actual human work.

Despite these various challenges, using models as metrics can provide nuanced and context-specific insights that are hard to capture with traditional evaluation methods. This approach should be applied with caution and rigorous methodology to ensure the most reliable and informative results.

## 3 How to make a statistically significant evaluation

We stressed out that the selection of appropriate evaluation metrics is critical. Equally critical is ensuring that the evaluation is statistically significant. For a robust evaluation, a sufficiently large test set is essential, along with the computation of score error bars. The figure 5 shows an example of a tight leaderboard from a past competition. In this particular competition, the third place candidate, qualified for a prize, only has a 0.0001 difference in score with the fourth place candidate. This thin margin between the third and fourth place highlights potential concerns regarding the evaluation methodology. This section presents methods for computing error bars, and addresses questions such as the ideal size of the test set. The goal is to provide methods to minimize the influence of randomness in competitions.

## 3.1 Error bars

Error bars are the representation of the uncertainty of a measurement, allowing to distinguish score estimations between candidate models. There are three common types of error bars: *standard deviation* (STD), *standard error of the mean* (SEM) and *confidence interval* (CI) (Krzywinski and Altman, 2013).

The **standard deviation** consists in the average distance between each sample and the mean:

$$\sigma = \sqrt{\frac{\sum_{i=1}^{n}(x_i - \bar{x})^2}{n-1}}$$

The use of $n-1$ in the denominator when computing the sample standard deviation is a result of what's called Bessel's correction (Bishop, 2006; Murphy, 2012). The main reason for using $n-1$ instead of $n$ is to provide an unbiased estimate of the population variance and standard deviation when computed from a sample.

The **standard error of the mean**, if the $n$ observations are statistically independents, is the standard deviation divided by the square root of the sample size:

$$SEM = \frac{\sigma}{\sqrt{n}}$$

The **confidence interval** is calculated by using the standard deviation to create a range of values likely – to a given probability (commonly 95%) – to contain the true population mean. This technique requires the computation of approximations in practice, and is not commonly used to analyze competitions and benchmarks in machine learning.

Given this context, what are the sources of variability in our area of interest? Specifically, in supervised learning, performance estimation often involves comparing model predictions with a test set's ground truth. Here, **the variability comes from the model in one hand, and the data in the other hand**.

The model can be stochastic during three different processes: the initialization, the learning process and the prediction process. Note that each of these processes is not necessarily stochastic in nature, and many models are completely deterministic. Even models that involve random initialization, such as neural networks, can be made deterministic by fixing a random seed[7]. This raises a question: should we impose to participants to fix a seed in order to reduce the variability of their methods? The main benefit of fixing the random seeds being improving the reproducibility of the methods. In general, having high variation of the results due to initialization can be a sign of low generalization capabilities. However, fixing the seed is controversial as it overlooks variability factors. In previous contests, we executed participants' code multiple times, choosing their poorest performance to motivate variance reduction. Although averaging multiple runs decreases variance, organizers shouldn't do this as it may favor high-variance methods; participants should ensure their methods have low variance.

The data is inherently stochastic, originating from real-world observations that can be considered as samples from unknown distributions. This randomness is compounded by variations in labeling quality, splitting into training and test sets, and other factors. One effective ways to mitigate these sources of variance is through high quantities of data. While it is often stated that the

---

7. A random seed is a number used to initialize a pseudo-random number generator, which is then used to generates the weights. A fixed random seed means that the number generator will always returns the same values.

quality of a machine learning model is largely determined by the quantity of available data, it is at least safe to say that abundant data enhances the reliability of model evaluations, as will be explored in subsequent sections.

The authors of Bouthillier et al. (2021) shows that most evaluations in deep learning focus on the impact of random weight initialization, which is only a small source of variance, comparable to residual fluctuations from hyperparameter optimization. However, this variance is much lower compared to the variance caused by splitting the data into training and test sets.



STATISTICAL HYPOTHESIS TESTING

Statistical hypothesis tests are used to decide whether the data at hand sufficiently support a particular hypothesis. In our area of interest, hypotheses often involve comparisons, such as whether algorithm *A* outperforms algorithm *B* or if the performance of various algorithms aligns with that of the baseline method. We mostly make multiple comparisons of multiple algorithms, multiple comparisons between two algorithms, or comparison between algorithms to a control (the baseline). The tests used for different scenarios are detailed in Japkowicz and Shah (2011). Even if the classical null hypothesis statistical tests (NHSTs) are widely used, recent research advocate for the use of Bayesian analysis instead (Benavoli et al., 2017). The authors present the Bayesian correlated *t*-test, the Bayesian signed rank test and a Bayesian hierarchical model that can be used for comparing the performance of classifiers, arguing that it solves the drawbacks of the frequentist tests. One of the drawbacks underlined is the fact that NHST computes the probability of getting the observed (or a larger) difference between classifiers if the null hypothesis of equivalence was true, which is not the probability of one classifier being more accurate than another, given the observed empirical results. Another common problem is that the common usage of NHST relies on the wrong assumptions that the *p*-value is a reasonable proxy for the probability of the null hypothesis (Demǎr, 2008). Other areas of science are also moving from NHSTs to Bayseian approaches, as evidenced by the journal *Basic and Applied Social Psychology*, which in 2015 banned the use of NHSTs and related statistical procedures (Trafimow and Marks, 2015).

In practice, in competitions, the statistical testing boils down to ranking participants and declaring ties. In the following, we examine the use of bootstrap and cross-validation as estimators of the variance in models performance.



BOOTSTRAP AND CROSS-VALIDATION

In the field of machine learning, *cross-validation* and *bootstrapping* are widely used ways of computing the bias and variance of models performances. These two methods are inherently different, as cross-validation involves re-training the model from scratch several times, while bootstrapping only uses re-shuffling of the test samples and predictions, making it quicker to compute. Validation methods help to prevent overfitting, a common issue in machine learning where a model performs well on the training data but poorly on new unseen data.

The **K-fold cross-validation** (CV) (Hastie et al., 2009) involves dividing the original data into several parts (folds), where one part is used for testing and the rest for training. This process is repeated multiple times, each time with a different part used for testing, and the performance metrics are averaged across all iterations to get a final evaluation of the model.

**Bootstrapping** (Efron and Tibshirani, 1993) involves generating multiple subsets of the original data set, using sampling with replacement, each having the same size as the original set. The

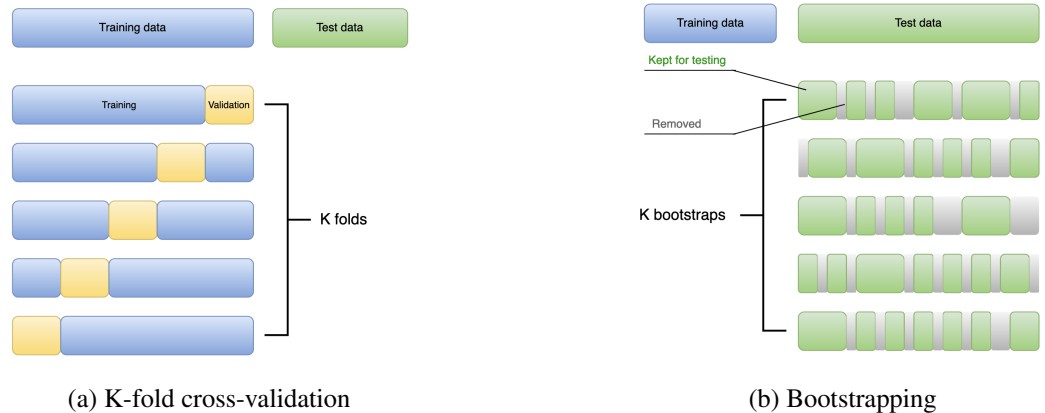

(a) K-fold cross-validation        (b) Bootstrapping

Figure 6: Schema of K-fold cross-validation (left) and bootstrapping (right). The cross-validation implies training on subsets of data with size determined by $K$. Bootstrapping involves sampling with replacement the test data, thus implying duplicated and missing samples in each evaluation. There is no limit to the number of re-sampling (bootstraps) that can be performed.

algorithm is then evaluated on each subset (known as a "bootstrap sample"). The performance metrics are averaged across all bootstrap samples to get a more robust evaluation of the algorithm.

K-fold CV and bootstrapping are illustrated and compared in the Figure 6. In both cases, the variance can be computed on the set of scores obtained. Note that, the bootstraps and the folds not being independents, we can't compute the standard error (dividing by the square root of the number of scores), as mentioned in Section 3.1.

The question of determining the best methods for estimating a model's generalization error received substantial attention in the literature, as evidenced by numerous studies (Nadeau and Bengio, 2003; Bengio and Grandvalet, 2004b; Markatou et al., 2005; Kohavi, 1995a; Zhang and Yang, 2015a; Dietterich, 1998; Tsamardinos et al., 2018; Esbensen and Geladi, 2010; Molinaro et al., 2005; Langford, 2005b,a; Forman and Scholz, 2010). This problem is deep and the suitability of an estimator appears to depend both on the evaluated models and the data they are evaluated on. Some empirical evaluations of generalization error estimators have been conducted (Kohavi, 1995b; Zhang and Yang, 2015b), and advise for a 10-fold CV. Top participants of the Performance Prediction Challenge (Guyon et al., 2006) used various cross-validation techniques to minimize average guess errors. The top performer employed virtual leave-one-out (VLOO) cross-validation for kernel classifiers, optimizing loss function through intensive cross-validation and using fresh data splits for re-estimation. While many preferred standard 10-fold cross-validation for hyperparameter tuning, others experimented with methods like bagging with bootstrap re-sampling or using challenge validation sets for predictions. Bengio and Grandvalet (2004a) demonstrate that, when dealing with simple cases, neglecting the dependencies between test errors can result in a bias that is roughly equal to the variance. These experiments highlight that one must exercise caution when interpreting the significance of differences in cross-validation scores.

Bootstrapping is more practical due to its computational efficiency and can be applied directly to results, eliminating the need to access the underlying algorithms (e.g. when evaluating results

submissions). Therefore, bootstrapping is highly valuable in the context of competitions and benchmarks, as it enables variance calculations without the need for computationally intensive operation.

## WITHIN GROUP AND BETWEEN GROUP VARIANCE

It is common to have multiple levels of granularity when computing scores and error bars. The highest granularity level is the level of data points, or samples. Samples can be grouped in lower granularity levels, such as tasks or datasets. Indeed, each task has a unique test set, leading to a distribution of scores for each algorithm on each task. This can also be defined as *within group variance*, the variance between the samples of a dataset, and *between group variance*, the variance between the tasks.

In this situation, the variance can be studied by invoking the law of total variance. The law of total variance (Weiss, 2005), also known as Eve's law, decomposes the variance of a random variable into two parts: the expected value of the variances conditioned on another random variable, and the variance of the expected values conditioned on that same random variable. Formally, let $X$ and $Y$ be two random variables. The law of total variance states:

$$\text{Var}(X) = \mathbb{E}[\text{Var}(X|Y)] + \text{Var}(\mathbb{E}[X|Y])$$

It naturally resonates with the concept of multi-granularity variance in the context of machine learning and algorithm evaluation. The overall variance in the performance of the algorithms (without considering tasks) is represented by $\text{Var}(X)$. The expected variance within each task (given the task) is similar to $\mathbb{E}[\text{Var}(X|Y)]$, where $Y$ denotes the specific task. The variance in the average performance of the algorithms across tasks is represented by $\text{Var}(\mathbb{E}[X|Y])$. It dissects the total variance of algorithm performance into parts: one due to inherent variability within each task, and another due to the variability in the algorithm's relative performance across different tasks. This view allows researchers and practitioners to understand the robustness of an algorithm across tasks and the variability of performance within specific tasks.

To dive more in-depth into this multi-level granularity scenario, we conducted experiments on the models' predictions from the AutoML (Guyon et al., 2019) and AutoDL (Liu et al., 2021) Challenges benchmarks. For each model, we estimated the within group and between group variance and compared the results. Our empirical experiments suggest that the highest granularity level exhibits lower variance of scores. Figure 7 shows the standard deviation of the ranks obtained by each participant of the AutoML and AutoDL challenges. In these challenges, the candidates are evaluated on a set of datasets, with a test set for each dataset. We can therefore compute the standard deviation of the ranks of each candidate, by varying the samples (high granularity) or the datasets (low granularity). This computation was performed using bootstraps, and highlights the difference in deviation depending on the granularity.

## 3.2 Size of test set

A crucial consideration is determining the test set size that provides a reliable error rate estimation. This should be the number one worry of the organizer: having enough test data to enable a robust judgement of the candidates. In the case of classification, Guyon et al. (1998) suggest as a rule of thumb to use $n = \frac{100}{p}$, where $n$ is the test set size and $p$ is the error rate of the best classifier, as estimated, for instance, by the human error rate. Hence, the better your classifier, the bigger your test set needs to be in order for you to compute precisely the error rates. In the case of imbalanced

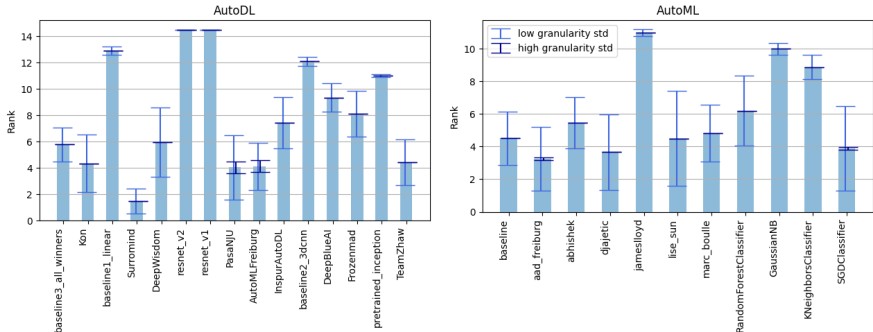

Figure 7: Average and standard deviation of ranks of AutoDL (left) and AutoML (right) candidates. The deviation is computed on bootstraps of data samples (high granularity) and bootstraps of datasets (low granularity).

classes, one can base this analysis on the size of the smallest group, or even regroup the least represented classes. More generally, outside of classification, the absolute precision on the scores or means can be used to separate the participants.

Recently, Guyon introduced a refined formula, which holds for all additive losses. The formula gives the sample size $n$ required to get a given precision $v$ and confidence $k$:

$$n = \frac{\sigma^2}{\mu^2} \times k^2 \times 10^{2v}$$

Where $\mu$ represents the mean error of the model evaluated, and $\sigma$ represents the standard deviation of the error rates. Interestingly, to increase the precision $v$ by one decimal, 100 times more test examples are needed. $k$, $\mu$ and $\sigma$ are squared, also indicating that the number of samples needed grows quickly under the influence of the error rate, the variance and the targeted confidence.

We highlight the importance of the number of test samples in an experiment conducted on tasks from the AutoDL Challenge (Liu et al., 2021). Only the datasets having less than 50,000 test samples were kept, for improved readability of the results. The experiment, done independently on each dataset, consist in increasing gradually the number of test samples used to compute the metrics and rank the candidate models. The size of the test set is therefore increased between 1 and $m$, $m$ being the total number of test samples of the dataset. For each intermediate value $i$, we sample with replacement (bootstrap) $i$ samples from the test set, and compute a ranking of the algorithms according to the scores obtained on this test set of $i$ samples. We perform $t = 5000$ trials of this procedure, resulting in $t$ different rankings, on which we compute the ranking stability using the Kendall $W$ concordance measure. The number of candidates $n$ is fixed in this experiment. $n$ should not have an impact on the value of the stability itself, but only the variance of this value.

The results, given in Figure 8, indicate that, unsurprisingly, the stability increase with the number of test samples. A value of stability of 1 means that the ranking of all methods does not change when bootstrapping the test samples. In this experiment, most datasets converge into a stability near 1 when the number of test samples reaches $10^4$. This indicates that the proposed models are well separated by the tasks. In the presence of ties, the stability converges to a value below 1. Under 1,000 samples, the rankings are unstable, meaning that there is an insufficient number of samples

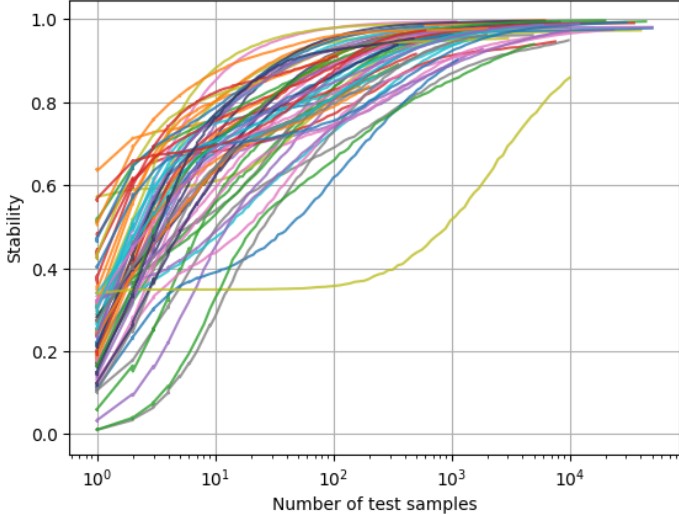

Figure 8: Evolution of the ranking stability depending on the number of test samples used to score the candidates. The stability is the Kendall $W$ measure computed on repeated trials. Each line is an independent dataset, and colors are displayed for readability.

to significantly rank the candidates by performance in the case of this benchmark. For benchmarks where the models are numerous and harder to separate, more than $10^4$ may be required in order to obtain a significant final ranking.

### 3.3 Avoid overfitting: staged evaluation

PUBLIC AND PRIVATE LEADERBOARDS

In order to do a reliable evaluation, you must divide the data in **at least** three sets: train, valida-tion and test. Note that validation and test sets are also commonly named "development and final phases"[8] or "public and private leaderboard". In some competitions, these sets contain distinct tasks or datasets; the validity of the argument still holds in these cases.

The **training set** is fundamental for model building. It includes both the features and the labels (i.e., the ground truth), which enable the model to learn the underlying patterns in the data. The **validation set** serves as an immediate check for how well the model is generalizing to unseen data. Although the ground truth is hidden, participants can get feedback on their performance. This en-ables them to tweak their models for improvement. It acts as a 'sandbox' for understanding how the model performs on data it hasn't seen before but could potentially overfit to if used improperly. The **test set** is the ultimate arbiter of a model's generalization capability. No feedback on performance is provided, preventing any last-minute tweaking that could artificially inflate the model's evaluation metrics. This is summarized in table 1. Ideally, these sets should share a similar data distribution, unless concept drift or shift is an inherent aspect of the problem being addressed.

---

8. Other synonyms of development phase include feed-back phase and practice phase.

The test set plays a critical role in this ecosystem by serving as a "firewall" against overfitting, since participants don't get feedback on their test set performance until the competition concludes. Indeed, receiving a repeated feedback from the leaderboard after each submission can lead participants to overfit their models to the validation data. The purpose of the test set is to evaluate performance on entirely new, unseen data, thereby ensuring that the winning solution is general rather than only excelling on the validation data. On a positive note, this empirical study (Roelofs et al., 2019), conducted on 120 Kaggle competitions, suggests that the overfitting between development and final phases (public and private leaderboards) is not common. This could either mean that most participants are adhering to best practices or that the dataset sizes and complexities are sufficient to mitigate the risks of overfitting.

The importance of splitting data into at least three sets — train, validation, and test — cannot be overstated for ensuring both the reliability and generalizability of machine learning models. This is even more crucial in competitive settings, such as machine learning competitions, where the temptation to fine-tune models based on leaderboard performance can potentially lead to overfitting. Some past competitions allowed participants to fine-tune their models during the final phase, which is generally not a good practice. It blurs the lines between validation and testing, compromising the integrity of the evaluation process. A well-structured competition should aim to measure a model's ability to generalize to new, unseen data, and letting participants fine-tune their models based on test set performance undermines this objective.

|  | **Train** | **Validation** | **Test** |
|---|---|---|---|
| Can participants access ground truth? | YES | NO | NO |
| Can participants obtain a score on it? | YES | YES | NO |
| Can organizers obtain a score on it? | YES | YES | YES |

Table 1: Train, validation and test sets. Here, the **validation set** refers to testing data hidden from participants; not to be confused with the validation procedure they can perform on the **train set**. The **test set** is for the final evaluation, avoiding leaderboard overfitting.

FILTERING PARTICIPANTS TO FINAL PHASE

While we do not declare the winner based on the validation set results in order to avoid "**participants overfitting**", this does not prevent another type of overfitting: "**organizer overfitting**".

The ambition of competitions is generally to recommend algorithms that could perform well on new tasks resembling that of the competition. Thus competitions are a problem in which the *organizers* perform a learning task: from the task(s) of the challenge, they select an algorithm that should perform well on new future tasks. Organizer overfitting occurs when the number of participants is large and rankings are noisy, increasing the chance of poorly selecting a winner. Competition organizers face a sad paradox: *the larger the number of participants, the more "successful" their competition*, but also *the greater the risk to overfit the particular competition setting*.

A heuristic often employed in sports, chess, and other types of competition is to use eliminatory trial runs to filter participants for the final competition phase. It has been highlighted that generalization can be improved by using the first phase of competition as a filter in machine learning competitions (Pavao et al., 2022b). The method simply consists in keeping only the *top-k* participants of the development phase into the final phase. However, determining the optimal *k*, the

number of top participants that we allow to access the final evaluation to optimize generalization, is hard to determine in practice.

A conservative choice of *k* that is preferable in practice, is to **eliminate participants who do not outperform the baseline methods provided by the organizers** with the "starting kit" (which may include well performing methods from previous challenges). This should be more acceptable to the participants than setting a hard threshold on the number of entrants of the *final phase*, and will at least eliminate the least serious participants who just submit the "starting kit".

## THE DANGERS OF SPLITTING

We have seen that splitting the data or tasks for training, validation and testing is necessary in order to evaluate participants fairly and avoid overfitting. It is common practice to completely shuffle randomly all data samples before splitting. This strategy assumes that data samples are independently and identically distributed across sets, avoiding to bias the evaluation towards one set or another. However, this approach can often yield misleading results, particularly in specialized domains where data naturally clusters into groups.

Consider a dataset composed of *n* microscopic images of cells collected from *m* different patients. These images aim to train a model for automated diagnosis, designed to generalize to new, previously unseen patients. If data splitting is executed at the image level, rather than the patient level, there's a high risk of overestimating model performance. In such cases, the model's apparent success in validation may not translate into effective generalization. This misleading effect is sometimes referred to as "voodoo machine learning" (Saeb et al., 2016). To better evaluate model performance in such scenarios, it's crucial to perform data splitting at the patient level, as illustrated in Figure 9, ensuring that all images from the same patient are grouped together in one of the training, validation, or testing sets.

The First Impressions Dataset (Escalante et al., 2018), utilized in various challenges, presents a similar data-splitting dilemma. The dataset consists in different videos of the same individuals, captured at different time intervals. to group all videos of the same individual into a single set—be it training, validation, or testing—to get a more accurate measure of the model's ability to generalize to new, unseen individuals. This approach reduces the risk of overfitting and better prepares the model for real-world applications.

More generally, all applications where the data are stratified in two or more levels must be split in a manner that respects these hierarchical structures to ensure accurate evaluation and robust generalization.

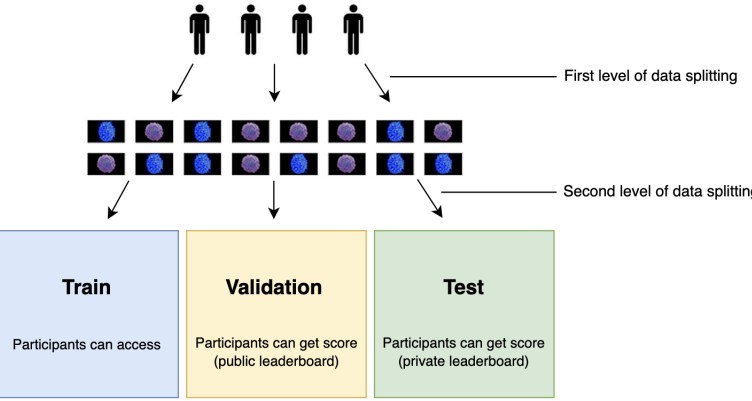

Figure 9: Illustration of several possible levels of data splitting. Here, the samples coming from the same source (the same person or element of the first level of splitting) should be kept in the same subsets of data to avoid overfitting.

## 4 How to fuse multiple scores

When judging and ranking the participants of a contest, we often need to combine the results from multiple criteria. This multi-score setting can emerge from different scenarios in machine learning competitions: when the models are tested on a set of tasks or datasets, when there are several evaluation trials (e.g. using cross-validation), or when multiple metrics or measures are used to rank the models. The case of having multiple heterogeneous metrics can typically arise when combining primary objective and secondary objective metrics when ranking candidate models. This problem, in a more general form, can be referred as the *problem of ranking* (Kendall and Smith, 1939): the goal is to rank a set of "candidates" $\mathscr{C}$, using the scores attributed to each of them by a set of "judges" $\mathscr{J}$. A judge here is simply any scoring procedure, hence producing a list containing one score for each candidate.

As each judges attributes a score to each candidates, the data of the problem can be represented by a score matrix $M$, as shown in Table 2. The problem then consists in using a ranking function $f : \mathbb{R}^{n \times m} \to \mathbb{R}^n$ to obtain a single ranking of candidates $\mathbf{r} = \texttt{rank}(f(M))$, with the function $\texttt{rank} : \mathbb{R}^n \to \mathbb{R}^n_+$ is defined as follows: $\forall i \in \{1, ..., n\}$, $\texttt{rank}(\mathbf{v})_i = 1 + \sum_{j \neq i} \mathbb{1}_{\mathbf{v}_j > \mathbf{v}_i} + \frac{1}{2} \sum_{j \neq i} \mathbb{1}_{\mathbf{v}_j = \mathbf{v}_i}$.

|  | Judge 1 | ... | Judge $m$ |
|---|---|---|---|
| Candidate 1 | $score_{11}$ | ... | ... |
| ... | ... | $score_{ij}$ | ... |
| Candidate $n$ | ... | ... | $score_{nm}$ |

Table 2: Score matrix. The judges can be of various nature (tasks, metrics, etc.). The output scores can also be of various types (real numbers, integers or ranks).

Table 3 shows an example of such problem. Depending on the ranking function $f$ chosen, the final ranking $\mathbf{r}$ will vary. Therefore, what is the good method to use? Intuitively, we want the final

ranking to represent as best as possible the opinions of all judges and to be congruent in this sense. However, this is a ill-defined objective. We can only propose methods that aim at answering this problem, and try to understand the underlying properties of the proposed methods.

|  | Judge 1 | Judge 2 | Judge 3 | Judge 4 |
|---|---|---|---|---|
| Candidate 1 | 0.8 | 0.5 | 0.7 | 0.5 |
| Candidate 2 | 0.6 | 0.9 | 0.4 | 0.5 |
| Candidate 3 | 0.4 | 0.7 | 0.8 | 0.5 |

Table 3: An example of a score matrix. A ranking function $f$ takes a score matrix as input and returns a **final ranking r** of the candidates.

We present the most common ranking functions in Section 4.1, their properties in Section 4.2 and give guidelines in Section 4.3.

## 4.1 Ranking functions

RANDOM DICTATOR

A straightforward approach to deriving a ranking from a matrix of scores involves uniformly selecting a judge at random and then adopting their judgment as the definitive ranking. This method is referred to as the *random ballot* or the *random dictator*. While it may initially seem counter-intuitive or even absurdly incorrect, it's astonishing how prevalent this method is in reality. In essence, the **random dictator is omnipresent**. Whenever we don't tackle a ranking problem head-on, but rather rely on a singular score to rank objects, we effectively permit the outcome to be governed by chance. This isolated score is typically drawn from a "mother distribution" and is consequently chosen at random. Examples of such scenarios include instances without re-runs, those lacking bootstrap re-sampling, or cases focusing on just one task. In such situations, the influence of the random dictator becomes evident.

MEAN AND MEDIAN

*Mean* and *Median* are average judges, obtained by either taking the mean (average value) or the median (middle value) over all judges, for each candidate. These two approach are fairly simple and very common in practice, especially the *mean*.

A potential issue with the *mean* is its sensitivity to extreme values, meaning that all judges don't have an equal impact on the outcome, especially if the scores are not normalized on the same scale, or are of different nature. The median leverage a bit this bias. On the other hand, if the scores are of similar nature, i.e. independently sampled from the same distribution, for examples several re-runs of the same experiment, then the *mean* naturally computes and converges to the expected value as the number of judges $m$ increases, as stated by the central limit theorem (Anderson, 2010) and the law of large numbers (Evans and J.S.Rosenthal, 2004).

AVERAGE RANK

*Average rank*, or *Borda count*, is defined as follows:

$$f(M) = \frac{1}{m} \sum_{\mathbf{j} \in \mathscr{J}} \texttt{rank}(\mathbf{j})$$

It has the interesting property of computing a ranking which minimizes the sum of the Spearman distance with all the input judges, as shown by Kendall and Gibbons (1990).

PAIRWISE COMPARISONS

*Pairwise comparisons* methods give scores based on comparisons of all pairs of candidates:

$$f(M) = \left( \frac{1}{(n-1)} \sum_{j \neq i} w(\mathbf{c}_i, \mathbf{c}_j) \right)_{1 \leq i \leq n}$$

where $w(\mathbf{c}_i, \mathbf{c}_j)$ represents the performance of $\mathbf{c}_i$ against $\mathbf{c}_j$. We can define different pairwise methods by designing different $w$ functions:

- *Copeland's method*: $w(\mathbf{u}, \mathbf{v}) = 1$ if the candidate $\mathbf{u}$ is more frequently better than the candidate $\mathbf{v}$ across all judges, 0.5 in case of a tie, and 0 otherwise.

- *Relative Difference*: $w(\mathbf{u}, \mathbf{v}) = \frac{1}{m} \sum_{k=1}^{m} \frac{u_k - v_k}{u_k + v_k}$.

In pairwise comparison methods, when a candidate beats all other candidates, it is a clear winner to be ranked first. If a candidate beats all other according to *Copeland's method*, it is said to be the *Condorcet winner*. However, there is no always a candidate that outplay all its opponents. This is because the majority preferences can be cyclic, thus exhibiting what is called a Condorcet paradox (Gehrlein, 1997). This property can be illustrated using Condorcet graphs, a graphical representation of pairwise comparisons between candidates. An arrow is drawn from one candidate to another when it performs better. Examples of such graphs, with a clear Condorcet winner, and exhibiting a cycle, are given in Figure 10.

OPTIMAL RANK AGGREGATION

*Optimal rank aggregation (ORA)* methods are a family of ranking methods that consist in proposing a distance function $d : \mathbb{R}^n \times \mathbb{R}^n \to \mathbb{R}_+$ and finding a ranking $r$ which minimizes the following objective function:

$$l(\mathbf{r}) = \sum_{\mathbf{j} \in \mathscr{J}} d(\mathbf{r}, \mathbf{j})$$

Some well-known distance functions that can be used are Kendall's $\tau$ distance, Spearman's distance or the Euclidean distance.

The ORA using Kendall's $\tau$ as a distance function is known as the *Kemeny-Young* method. It has interesting properties such as being a Condorcet method and satisfying Local IIA (defined below); however, its computation is NP-Hard. The high complexity of the *Kemeny-Young* method prevented us from including it in the experiments.

The ORA using the Spearman distance also has interesting properties and is computationally linear as it produces the same ranking as the *average rank* method (Kendall and Gibbons, 1990), as mentioned earlier.

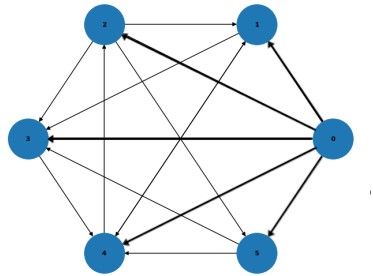

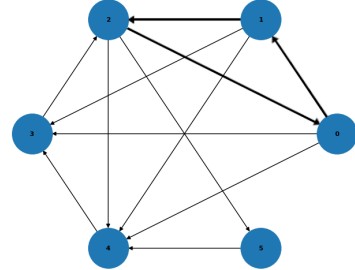

(a) *Candidate 0* is the Condorcet winner.      (b) There is no Condorcet winner.

Figure 10: Condorcet graphs where vertices represents candidates, and where there is an arrow between candidates **u** and **v** if **u** is more frequently better than **v** according to the judges' preferences. In the left example, *Candidate 0* is a clear Condorcet winner, beating all other candidates. The right example exhibit a Condorcet paradox, as *Candidate 0* beats *Candidate 1*, *Candidate 1* beats *Candidate 2* and *Candidate 2* beats *Candidate 0*, resulting in a cycle. Bold arrows are highlighted for clarity.

In practice, the optimization can be performed using differential evolution (Storn and Price, 1997). A good overview of ORA and rank distance functions is given in Heiser and D'Ambrosio (2013).

## 4.2 The problem of ranking is not trivial

No ranking function perfectly captures the judges preferences when there are more than two candidates. This idea is well highlighted by an important result from social choice theory: Gibbard's theorem (Allan, 1973), a generalization of Arrow's theorem (J., 1950).

**Theorem 1.** *Gibbard's theorem. Any deterministic ranking method holds at least one of the following three (unwanted) properties:*

1. *The process is dictatorial[9],*

2. *The ranking is limited to only two candidates,*

3. *The process is open to "tactical voting": the preferences of a judge may not best defend their interest.*

In practice, this imply incompatibilities between several desired properties of ranking methods. Some of the theoretical properties satisfied or not by the methods defined here are summarized in Table 4. These properties are defined below.

---

9. In a dictatorial process, a single judge can fully dictate the outcome.

| | Winner | | Judge perturbation | | Candidate perturbation | | |
|---|---|---|---|---|---|---|---|
| | Majority | Condorcet | Consistency | Participation | IIA | LIIA | Clone-proof |
| Random | | | ✓ | ✓ | ✓ | ✓ | ✓ |
| Mean | | | ✓ | ✓ | ✓ | ✓ | ✓ |
| Median | | | | | ✓ | ✓ | ✓ |
| Average rank | | | ✓ | ✓ | | | |
| Copeland | ✓ | ✓ | | | | | |
| Kemeny-Young | ✓ | ✓ | | | | ✓ | |

Table 4: Main properties satisfied or not by the ranking functions.

**Majority criterion** (Rothe, 2015): If one candidate is ranked first by a majority (more than 50%) of judges, then that candidate must win.

**Condorcet criterion**: The Condorcet winner is always ranked first if one exists. The Condorcet winner is the candidate that would obtain majority against each of the others when every pair of candidates is compared. The Condorcet criterion is stronger than the Majority criterion.

**Consistency**: Whenever the set of judges is divided (arbitrarily) into several parts and rankings in those parts garner the same result, then a ranking on the entire judge set also garners that result.

**Participation criterion**: The removal of a judge from an existing score matrix, where candidate $u$ is strictly preferred to candidate $v$, should only improve the final position of $v$ relatively to $u$.

**Independence of irrelevant alternatives (IIA)**: The final ranking between candidates $u$ and $v$ depends only on the individual preferences between $u$ and $v$ (as opposed to depending on the preferences of other candidates as well).

**Local IIA (LIIA)** (weaker): If candidates in a subset are in consecutive positions in the final ranking, then their relative order must not change if all other candidates get removed.

**Independence of clones (clone-proof)**: Removing or adding clones of candidates must not change the final ranking between all other candidates.

## 4.3 Guidelines

We have presented different ranking functions, and learned that none of them satisfies all the desired theoretical properties mentioned above. If some of these theoretical properties are absolutely required in your benchmark or competition, then they can dictate the function to chose. However, in most cases, this theoretical analysis shade some light but does not settle the problem once for all. To give insights in practical and machine learning related scenarios, (Brazdil and Soares, 2000) and later (Pavao et al., 2021b) have conducted empirical studies, comparing ranking functions using empirical criteria and running experiments on machine learning benchmarks. The results points that the *average rank* method fares well, in terms of meta-generalization and stability.

*Weighted average rank*, in which we attribute a weight to each judge before computing the average, can also be a good choice when there is a clear difference in the importance of each judge in the evaluation. The *weighted average rank* function can be expressed as following, with **w** the list of weights:

$$f(M) = \frac{1}{m} \sum_{\mathbf{j} \in \mathcal{J}} w_j \times \mathtt{rank}(\mathbf{j})$$

## 5 Conclusion

This chapter offers a categorization of evaluation metrics, covering performance, ethics and societal impact, resource utilization, and evaluator viewpoints (human or model based evaluation). It is clear that real-world constraints must be a part of any evaluation, as they ensure that the results are practical and applicable. To ensure statistically reliable evaluations, it is recommended to have a consequent test set and to use bootstrap methods to assess error bars, given their computational efficiency and accuracy. The required number of test set samples grows quadratically with the mean error, the standard deviation of error rates and the confidence, while it grows exponentially with the targeted precision level.

We also stressed out the importance of having a distinct final phase prevents potential overfitting. Filtering participants accessing the final phase improve the generalization of the winner selection (Pavao et al., 2022a). To enhance the relevance of results, as a rule of thumb, only participants that exceed a predefined baseline should be allowed into this phase.

Finally, we studied the methods for aggregating results from multiple scores. The average rank method showed its efficiency (Pavao et al., 2021a), while remaining simple to compute and to interpret.

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
