# OpenReview forum: "How to judge a competition: Fairly judging a competition or assessing benchmark results"
_DMLR — Accepted by DMLR_

### Review · Reviewer_HDdA · 2024-08-30

**Recommendation:** 4
**Confidence:** 2

**Summary Of Contributions:**

This submission presents a comprehensive approach to judging the competitions. It introduces a categorization of evaluation metrics, the ways of making the statistically significant evaluation, and aggregating the scores.

**Strengths:**

- **Meaningful Topic**: The topic of this paper is very important. Most AI research takes place in competitions, but there aren't many references that discuss the methodology of evaluation in these settings. If researchers aren't well-trained in evaluation methods, the conclusions of their papers might not be reliable. I believe that summarizing and studying this issue is very valuable.
- **Good Coverage**: The paper does a good job of covering different metrics. It discusses a wide range of metrics, many of which align with my own knowledge.
- **Practical Methodology**: The methodology and aggregation methods discussed in the paper are logical and well thought out. I believe they will be helpful to others in the field and provide useful guidance for evaluating AI research.

**Audience:**

Yes

**Broader Impact Concerns:**

No concern.

**Claims And Evidence:**

Yes.

**Datasets And Benchmarks:**

N/A

**Extended Submissions:**

No.

**Limitations:**

These points might not be weaknesses that need improvement, as the current submission is already clear and thorough. I’m interested in the following related topics that haven't been covered in the paper:

- **Generalization and Application**: The paper discusses various applications and categories of evaluation metrics. I would have appreciated more examples and in-depth analysis showing how these methodologies can be generalized and applied across different areas. This would help readers understand the broader applicability of the proposed methods in diverse fields.
- **Challenges in Evaluating Large Language Models (LLMs)**: The competition in recent research related to Large Language Models (LLMs) and their applications, such as LLM agents, also faces significant challenges in fair judgment. There are various benchmarks and tasks being developed for evaluating these models, but the situation appears to go beyond just fair evaluation. How the proposed methodologies might address these issues or if they need to be adapted for LLMs.
- **Use of LLMs as Evaluation Methods**: LLMs are increasingly being used as evaluation tools themselves. It would be valuable to have a discussion on this emerging methodology. What are the potential benefits and drawbacks of using LLMs for evaluation, and how might this influence the fairness and reliability of competition results?

**Requested Changes:**

N/A

**Strengths And Weaknesses:**

This chapter is well-structured and easily to follow. The discussion on each subtopics are thorough and clear, which I think is good for a chapter of a book. The coverage of the contents is also comprehensive, which aligns my experience and prior knowledge.

---

### Review · Reviewer_xYG5 · 2026-04-15

**Recommendation:** 3
**Confidence:** 2

**Summary Of Contributions:**

This submission provides a practical framework for minimizing randomness and ensuring fairness when evaluating machine learning competitions: it categorizes evaluation metrics into four main areas: performance, ethical/societal impact, resource consumption, and evaluator-centric approaches, and details how to compute error bars and determine adequate test set sizes. To prevent both participant and "organizer" overfitting, the author advocates for strict staged evaluations using separate train, validation, and test sets.

**Strengths:**

1. The submission provides a highly valuable and comprehensive framework aimed at minimizing randomness and uncertainty in the evaluation of machine learning competitions.
2. It successfully translates concepts from social choice theory, specifically Gibbard's theorem and Arrow's theorem, directly into the context of ranking machine learning models.
3. It introduces critical, actionable concepts such as identifying and preventing "organizer overfitting" and "participant overfitting" on public leaderboards.

**Audience:**

Yes

**Broader Impact Concerns:**

This paper provides a practical framework for ensuring fair and statistically rigorous evaluations in machine learning competitions. It guides organizers on minimizing randomness by selecting appropriate evaluation metrics, calculating necessary test set sizes, preventing overfitting through staged evaluations, and fairly aggregating multiple participant scores.

**Claims And Evidence:**

The claims are generally reasonable, but several key aspects lack sufficient justification and rigor. Please refer to the Requested Changes section for detailed concerns.

**Datasets And Benchmarks:**

It does not introduce a dataset or a new benchmark for the community to use.

**Extended Submissions:**

It is not an extended submission.

**Limitations:**

The submission would be significantly strengthened by validating its variance and stability claims (e.g., Figure 8) against datasets from disparate domains, such as natural language generation or generative adversarial networks.

**Requested Changes:**

1. Lack of justification and citation for the test set size formula in Section 3.2.
The manuscript introduces a formula for estimating test set size without providing a reference or derivation. It briefly states that “Guyon introduced a refined formula, which holds for all additive losses,” but does not justify this claim. This is scientifically insufficient. The authors should (i) include a concise mathematical derivation or proof explaining why the formula applies specifically to additive loss functions, and (ii) provide a formal academic citation for this result. Without such justification, benchmark organizers cannot reliably apply the proposed formula.

2. Missing discussion of mitigation strategies for LLM-as-a-judge bias.
Section 2.4 highlights an important issue: ChatGPT tends to produce overly optimistic Likert scores and exhibits self-preference bias when evaluating its own outputs. Given the increasing adoption of LLM-as-a-judge paradigms in benchmark design, the paper would benefit from a dedicated discussion of practical mitigation strategies for this “circular reasoning” problem. For example, the authors could consider anonymizing generation artifacts, using ensembles of heterogeneous LLMs, or applying controlled prompting and temperature settings to reduce self-preference bias.

3. Insufficient mathematical formalism in the RL subsection.
The reinforcement learning subsection currently remains largely qualitative. While it correctly notes the need for multiple trials with different random seeds and references “behavioral metrics,” it lacks formal definitions. The authors should strengthen this section by introducing precise mathematical formulations for at least two standard RL evaluation metrics (e.g., expected return, success rate, or regret).

**Strengths And Weaknesses:**

Strengths:
1. It effectively groups metrics into performance, ethical and societal impact,  resource consumption, and evaluator-centric approaches.
2. The paper clearly translates social choice theory and Condorcet paradoxes into the context of ranking machine learning models.
3. By comparing methods like the Random Dictator, Mean, Copeland's method, and ORA against theoretical properties (Majority, IIA, etc.), it provides a robust justification for its final recommendation to use the Average Rank method.
Weaknesses:
1. The paper's claims are interesting, but certain sections lack the necessary mathematical and empirical depth.  Specifically, the RL section notes that evaluations require "multiple trials with different random seeds" and generic "behavioral metrics," but completely misses the mathematical rigor, specific formulas, and concrete metric recommendations that made the classification section so effective.

---

### Review · Reviewer_QEgg · 2026-05-16

**Recommendation:** 4
**Confidence:** 3

**Summary Of Contributions:**

This chapter provides a comprehensive framework and practical guidelines for minimizing randomness and ensuring fairness when judging machine learning competitions and benchmarks. The author explores a wide taxonomy of scoring metrics, including performance metrics, ethical and societal impact metrics, resource consumption metrics, and evaluator-centric approaches. Furthermore, the work addresses the statistical significance of evaluations by discussing error bars, the necessary size of test datasets, and strategies to prevent overfitting through staged evaluation. Finally, the paper investigates methods for score aggregation, detailing various ranking functions and their theoretical properties.

**Strengths:**

The primary strength of this submission lies in its dual nature: it is both theoretically grounded and immensely practical.

- The author effectively uses empirical data from past competitions, such as the AutoDL and AutoML challenges, to visually and statistically demonstrate variance and ranking stability.
- The inclusion of a metric like Adversarial Accuracy ($AA_{TS}$) to quantify privacy leakage in generative models is a strong addition that addresses modern AI evaluation concerns.
- The author adeptly handles the complex issue of rank aggregation by invoking Gibbard's theorem, clearly explaining to practitioners why no perfect ranking system exists and offering the average rank method as a solid empirical compromise.

**Audience:**

Yes

**Claims And Evidence:**

yes

**Datasets And Benchmarks:**

The paper does not introduce a new dataset or benchmark; rather, it analyzes existing ones (e.g., Global Wheat Detection, AutoDL, First Impressions Dataset) to illustrate evaluation principles. The citations provided in the references section are sufficient for readers to locate the original benchmark data.

**Extended Submissions:**

NA

**Limitations:**

The paper focuses heavily on traditional structured competitions and relies on historical benchmark data (e.g., NIPS 2003 Feature Selection, AutoML 2015-2018). While these are foundational, the landscape of AI benchmarks is rapidly shifting toward evaluating open-ended generative models, which introduces subjectivity that is difficult to quantify using the standard test-set sizing formulas provided.

**Requested Changes:**

I'd recommend work on the following fronts:

- Expanding the discussion on unsupervised learning metrics, specifically addressing how to benchmark large language models (LLMs) beyond the brief mentions of human-centric and model-centric approaches, would strengthen the work.
- The paper notes that continuous-state RL problems struggle with state abstractions, but could benefit from explicitly detailing one or two standard behavioral metrics used in modern RL challenges.

**Strengths And Weaknesses:**

## Strengths:
- The paper provides a highly structured and extensive overview of metrics, moving beyond simple accuracy to include crucial ethical, societal, and resource-based evaluations.
- It offers highly practical mathematical tools for organizers, such as Guyon's formula for determining the required test set size based on desired precision and confidence.
- The discussion on data splitting is insightful, particularly the warnings against "voodoo machine learning" when hierarchical data structures (like multiple images from a single patient) are improperly shuffled.

## Weaknesses:
- While the classification and regression sections are robust, the coverage of unsupervised learning and reinforcement learning metrics feels comparatively brief and could benefit from more detailed practical examples.
- The optimal rank aggregation (ORA) section notes that the Kemeny-Young method is NP-Hard, but does not deeply explore approximation algorithms that organizers might use in practice.